# Computational Approaches to Enzyme Inhibition by Marine Natural Products in the Search for New Drugs

**DOI:** 10.3390/md21020100

**Published:** 2023-01-30

**Authors:** Federico Gago

**Affiliations:** Department of Biomedical Sciences & IQM-CSIC Associate Unit, School of Medicine and Health Sciences, University of Alcalá, E-28805 Madrid, Alcalá de Henares, Spain; federico.gago@uah.es; Tel.: +34-918854514

**Keywords:** enzyme inhibitors, databases, cheminformatics

## Abstract

The exploration of biologically relevant chemical space for the discovery of small bioactive molecules present in marine organisms has led not only to important advances in certain therapeutic areas, but also to a better understanding of many life processes. The still largely untapped reservoir of countless metabolites that play biological roles in marine invertebrates and microorganisms opens new avenues and poses new challenges for research. Computational technologies provide the means to (i) organize chemical and biological information in easily searchable and hyperlinked databases and knowledgebases; (ii) carry out cheminformatic analyses on natural products; (iii) mine microbial genomes for known and cryptic biosynthetic pathways; (iv) explore global networks that connect active compounds to their targets (often including enzymes); (v) solve structures of ligands, targets, and their respective complexes using X-ray crystallography and NMR techniques, thus enabling virtual screening and structure-based drug design; and (vi) build molecular models to simulate ligand binding and understand mechanisms of action in atomic detail. Marine natural products are viewed today not only as potential drugs, but also as an invaluable source of chemical inspiration for the development of novel chemotypes to be used in chemical biology and medicinal chemistry research.

## 1. Overview

Both pharmacology and basic cell biology have traditionally benefited from the continuous identification and biochemical characterization of active principles obtained from natural sources. The scanty primitive chemical libraries of natural products (NPs), consisting mostly of the alkaloids and heterosides isolated from terrestrial plants that provided the foundations of modern pharmacology [1], were progressively enriched with a multitude of small- to medium-sized molecules present in numerous living creatures, both big and small, including those inhabiting seas and oceans, which together make up a huge water mass that covers >70% of Earth’s total surface and hosts ~80% of all living species [2]. Nonetheless, and despite a notable renaissance in recent years [3,4], the list of marine natural products (MNPs) that have been approved or are currently found in the global marine pharmaceutical clinical pipeline (https://www.midwestern.edu/departments/marinepharmacology/clinical-pipeline, accessed on 20 December 2022) is still very limited, and only a few of these drugs actually target an enzyme.

The vastness of the largely unexplored chemical space [5] existing in marine environments poses daunting challenges in terms of (i) sample recollection, (ii) compound isolation, (iii) chemical characterization, (iv) evaluation in as many biochemical and/or biological assays as possible, preferably using validated targets and high-throughput state-of-the-art technologies [6], and (v) the identification and validation of pharmacologically relevant targets. Given the precedents of successful marine leads as a source of useful medicinal agents and biochemical probes, it can be argued that it makes sense to continue exploiting over four billion years of evolution in nature’s combinatorial chemistry, often subjected to unique ecological pressures and nutrient availability, that led to selective survival advantages in the producing organisms [7]. The best studied phylogenetically diverse living beings from marine habitats include green, brown, and red algae; sponges; coelenterates (i.e., jellyfishes, corals, and sea anemones); bryozoans (i.e., invertebrates known as moss animals); the Ascidiacea class (commonly known as the ascidians, tunicates, and sea squirts); mollusks; echinoderms; phytoplankton; and innumerable bacteria and fungi. Secondary metabolites are specialized organic compounds that are not considered essential for normal growth or reproduction (under laboratory culture conditions) but instead play roles in evolution, communication (as chemical cues), and competition, or else appear to be used as chemical weaponry against prey or natural enemies in their natural environments. MNPs often feature unique scaffolds and carbocyclic skeletons, and many have been discovered following their bioassay-guided isolation, although the paucity of material usually prevents the full profiling of bioactivity [8], which is often limited to some rudimentary tests (e.g., phenotype-oriented antimicrobial or cytotoxic assays [9], and/or inhibitory activity against one enzyme or a limited set of enzymes). In this respect, it has been pointed out that micromolar activities detected in extracts should be critically analyzed because of potential artefactual assay readouts due to unspecific aggregation [10], hence the recommendation to use β-lactamase and malate dehydrogenase as counter-screening enzymes [11], among other precautionary measures.

Recent progress in understanding the genetic basis of MNP biosynthesis and the ever-increasing availability of genomic information have created unique opportunities to develop sequence-based approaches for the discovery of novel bioactive molecular entities [12]. Polyketide synthases (PKSs) and multimodular nonribosomal peptide synthetases (NRPS) stand out among the enzymes that are ultimately responsible for the highly efficient synthesis of three large subclasses of important NPs (PKs, NRPs, and PK/RP or NRP/PK hybrids) [13] through the concerted assembly of relatively simple carboxylic acid and amino acid building blocks, respectively [14,15]. Type I PKSs consist of multiple modules, with each module minimally containing three core domains: acyltransferase (AT) domain, ketosynthase (KS) domain, and thiolation (T) domain [aka acyl carrier protein (ACP) domain] [16]. These (mega)enzymes are encoded in biosynthetic gene clusters (BGCs), which have been identified for hundreds of bacterial and fungal metabolites and are highly evolved for horizontal exchange [17]. Besides, attention continues to be drawn to two facts that have significantly expanded the area of MNP research, namely (i) that some isolated MNPs are bioaccumulated in the target organism from dietary sources, e.g., algae [18]; and (ii) that a significant number of MNPs are actually produced by microbes and/or microbial interactions with the “host from whence it was isolated” [8]. The growing emphasis on the study of compounds from microbial sources (both terrestrial and marine) has been fueled by interest in (i) the central role that microorganisms play in mediating both interspecies interactions and host-microbe relationships [19]; and (ii) their natural ability to produce ribosomally synthesized and post-translationally modified peptides (RiPPs), which often contain noncanonical amino acids and structural motifs that give rise to a currently under-represented class of biologically active molecules [20,21].

Modern science (and the world at large) is overly dependent on computer and internet technologies. Computers have a long history in data management, as well as in information storage, processing, retrieval, and dissemination, and for these purposes their use has expanded enormously in recent years and has contributed to shaping the current research landscapes in bioscience and biomedicine as we know them today. The World Wide Web has become a central source of (i) information on all possible subjects that is stored and (ideally) curated in extensively hyperlinked databases; (ii) educational and research tools; and (iii) services that are intended to make life easier not only for the general public, but also for scientists, including those devoted to chemical biology, medicinal chemistry, and drug discovery. Devices ranging from pocket computers, also known as mobile or cellular phones which have superseded earlier personal digital assistants (PDA), to tablets, laptops, desktops, mainframes, and supercomputers dominate many aspects of our lives and complement human skills in numerous applications designed to utilize an ever-growing torrent of biological and chemical data in effective manners. While this is the driving force behind the increasing use of high-performance computing, machine learning and artificial intelligence for processing tons of data in a way that compensates for the inherent constraints of human cognition [22], better-informed decision making in drug discovery and development still largely relies (or so I like to believe) on the power of human judgement and life-long expertise. 

The concise Guide to Pharmacology (https://www.guidetopharmacology.org/; latest release 13 October 2022, accessed on 20 December 2022) presented by the International Union of Basic and Clinical Pharmacology (IUPHAR) and the British Pharmacological Society (BPS) includes enzymes (Nature’s catalysts essential to the chemistry of life) as one of the six major classes of pharmacological targets, the others being G protein-coupled receptors, ion channels, nuclear hormone receptors, catalytic receptors, and transporters (including the very large SLC superfamily of solute carriers) [23]. Over one thousand distinct human enzymes are described in the Universal Protein Knowledgebase (UniProtKB) [24], therefore representing almost half of all current human targets. Fortunately, the three-dimensional (3D) structures of many of these enzymes or closely related counterparts from other species—both in their apo forms and in complexes with ligands—have been solved and deposited in the Worldwide Protein Data Bank (wwPDB) [25], a continuously enlarging global repository established in 1971. These 3D structures facilitate the elucidation of functional mechanisms, aid in understanding the binding mode of inhibitors, and enable virtual screening (VS) and structure-based drug design (SBDD) technologies. For other enzymes of interest, we still depend on several homology modeling approaches [26], neural network-based models, such as those generated by AlphaFold [27], and artificial intelligence, which was recently employed to build the ESM Metagenomic Atlas (https://esmatlas.com/, accessed on 20 December 2022), with more than 617 million structures from all kingdoms of life [28].

Following the recommendations of the Nomenclature Committee of the International Union of Biochemistry and Molecular Biology (IUBMB, https://www.qmul.ac.uk/sbcs/iubmb/enzyme/; accessed on 20 December 2022), the wwPDB assigns Enzyme Commission (EC) numbers to protein chains in macromolecular structures according to the type of chemical reaction that they catalyze. The main classes are oxidoreductases (EC 1), transferases (EC 2), hydrolases (EC 3), lyases (EC 4), isomerases (EC 5), ligases (EC 6), and translocases (EC 7), with subclasses (with up to 4 digits) being defined on the basis of the specific donors and receptors of chemical groups that participate in the reactions and additional considerations. The main collection of functional enzyme and metabolism data is possibly BRENDA (https://www.brenda-enzymes.org/, accessed on 20 December 2022), which was established in 1987 and selected as an ELIXIR Core Data Resource [29] in 2018 [30]. In addition, the merging of MACiE (Mechanism, Annotation and Classification in Enzymes), a database of enzyme mechanisms, and CSA (Catalytic Site Atlas), a database of catalytic sites of enzymes, has resulted in the M-CSA Mechanism and Catalytic Site Atlas (http://www.ebi.ac.uk/thornton-srv/m-csa/browse/?sort=ec, accessed on 20 December 2022) [31], which consolidates a body of knowledge on enzyme structures, gene sequences, reaction mechanisms, metabolic pathways, and kinetic data that any researcher working on enzyme inhibitors should be familiar with. 

Building on this introductory background information, the following sections will separately cover each of the abovementioned aspects for which specialized computer technologies have been developed in the field of enzyme inhibition by MNPs (Figure 1).

## 2. Bibliographical Sources and Virtual NP Databases

Chemical libraries encompassing millions of compounds include the Chemical Abstracts Service (CAS) REGISTRY database (http://www.cas.org/expertise/cascontent/registry/index.html, accessed on 20 December 2022), which is updated on a daily basis and contains >250,000 NPs out of >150 million chemical substances, PubChem (including PCSubstance, PCCompound, and PCBioAssay) [32], ChEMBL (a manually curated database of >2,300,000 bioactive molecules with drug-like properties, last update July 2022) [33], and ChemSpider (with various levels of partial to complete stereochemistry) [34]. The free-to-access resource DrugBank is a web-enabled database (https://go.drugbank.com/, accessed on 20 December 2022) that incorporates comprehensive molecular information about drugs, their mechanisms, their interactions, and their targets. First described in 2006 as a knowledgebase for drugs, drug actions, and drug targets [35], DrugBank has evolved over time in response to improvements in web standards and changing needs for drug research and development. The latest update, DrugBank 5.0 [36], was expanded to cover not only drug binding data, numerous investigational drugs, drug-drug and drug-food interactions, and SNP-associated drug effects, but also information on the influence of hundreds of drugs on metabolite levels (pharmacometabolomics), gene expression levels (pharmacotranscriptomics), and protein expression levels (pharmacoproteomics). Enzyme inhibitors (DBCAT000003) are described as “compounds or agents that combine with an enzyme in such a manner as to prevent the normal substrate-enzyme combination and the catalytic reaction”.

Reviews on MNPs have been published on a regular basis in the scientific literature [37,38,39]. The renewed upsurge of interest in NPs, and MNPs in particular, over the last two decades has led to a rapid multiplication of databases in both the private sector and the public domain that compile general-purpose or thematic information on these naturally occurring compounds, often incorporating supplementary material published in scientific papers. A dedicated, searchable, and continuously updated database (MarinLit, https://marinlit.rsc.org/, accessed on 20 December 2022) that was established in the 1970s by Prof. John Blunt and Prof. Murray Munro (University of Canterbury, New Zealand) has been maintained by the Royal Society of Chemistry (UK) since 2014. MarinLit covers ~40,000 compounds from marine macro- and microorganisms and about the same number of references to journal articles. Among the specialized MNP databases, the Dictionary of Marine Natural Products (DMNP) [40] appeared as the first of its kind in 2008 and encompassed a subset of data from the Dictionary of Natural Products (DNP, one of several Chapman & Hall chemical dictionaries) based on the biological source of the compounds. DMNP was marketed as a book together with a CD-ROM for a desktop version, and the searchable web-based version CHEMnetBASE (https://dmnp.chemnetbase.com/, accessed on 20 December 2022) is still available (v. 31.1; updated in 2022), but only to subscribing institutions.

Virtual chemical libraries of NPs can be categorized into (i) encyclopedic and general NP databases; (ii) special subsets within fully enumerated, ultra-large scale chemical libraries specifically built to facilitate VS campaigns, e.g., ZINC [41,42]; (iii) compound collections enriched with NPs used in traditional medicines; and (iv) specialized databases focused on specific habitats, geographical regions, organisms, biological activities, or even specific NP classes. Unfortunately, many NP databases belonging to the latter two categories are rather ephemeral or rapidly become either outdated or unavailable to the scientific community [43], and the same criticism applies to many bioinformatics web services related to NPs [44]. This is most likely due to (i) a lack of funds (and/or human resources) for their sustained management and continuous upgrading, and (ii) the current overwhelming “data deluge”. For these reasons, there is an urgent need for nonredundant, community-wide efforts that optimize the use of contemporary bioinformatic and chemoinformatic capabilities, as exemplified by the recently established open platform LOTUS (https://lotus.naturalproducts.net, accessed on 20 December 2022), a knowledgebase that is expected to have strong transformative potential for research on NPs and beyond [45]. In this praiseworthy initiative, data sharing within the Wikidata framework broadens interoperability and facilitates access to >750,000 referenced structure-organism pairs.

Another large and freely available NP database is Super Natural II (https://bioinf-applied.charite.de/supernatural_new/index.php; last updated: October 2022, accessed on 20 December 2022), which provides two-dimensional (2D) structures and physicochemical properties for ~326,000 molecules, as well as information about the pathways associated to their synthesis, degradation, and mechanisms of action with respect to structurally similar drugs [46]. An additional recent compilation of 400,000 non-redundant NPs was made available in 2021 [47] as the open-access COlleCtion of Open NatUral producTs (COCONUT, https://coconut.naturalproducts.net/, accessed on 20 December 2022).

One important goal of these NP databases is to facilitate a quick assessment of novelty for any newly identified compound in a natural extract. To distinguish between known and unknown compounds, it is important to have rapid and trustworthy “dereplication” methods, which rely heavily on the interpretation of molecular mass and molecular formula, as well as UV and NMR spectral data [48]. Nevertheless, the dereplication process can be problematic sometimes because (i) the present validity and accuracy of the collected information is only as good as that of the original data source; and (ii) stereochemical information on NPs is often inaccurate or incomplete. In the field of MNPs alone, it was recently reported that more than 200 structures were misassigned in the last ten years only [49]. A comparative analysis of the original and the revised structures revealed that major pitfalls still plague the structural elucidation of small molecules and, consequently, that quite a few 3D molecular structures present in databases may be inaccurate. This finding emphasizes the roles of total synthesis, X-ray crystallography, as well as chemical and biosynthetic logic, to complement spectroscopic data. Nevertheless, it is noteworthy that a much lower incidence of “impossible” structures was found in MNPs compared to NPs of plant origin.

The utilization of computer-assisted structure elucidation (CASE) programs can minimize the risk of misassignment and help identify truly novel compounds (the “unknown unknowns”) [50] by generating all structures that are consistent with key data from 2D correlation spectroscopy (COSY), heteronuclear multiple bond correlation (HMBC), and 1,1-adequate sensitivity double-quantum spectroscopy (ADEQUATE) NMR experiments, and by ranking the resulting structures in order of probability. The algorithms may additionally benefit from both stereospecific NMR data and use of optimized geometries and predicted chemical shifts provided by density funtional theory (DFT) quantum mechanical calculations [51]. The absolute configuration of an MNP can be unequivocally confirmed by crystallographic analysis and, in the case of noncrystalline compounds containing a pseudo-meso core structure that results in a specific rotation ([a]_D_) of almost zero (e.g., elatenyne), it may be necessary to absorb the compound into a porous coordination network (a “crystalline sponge”) [52].

The exploration of the identities and biological activities of metabolites present in complex mixtures has benefited enormously in recent years from scalable native and functional metabolomics approaches [53]. Novel techniques, such as affinity selection mass spectrometry (MS), complemented with pulsed ultrafiltration, size exclusion chromatography, and magnetic microbead affinity selection screening, now allow the separation of non-covalent ligand-receptor complexes from other nonbinding compounds [54]. 

Recognizing the need for community-wide platforms to effectively share and analyze raw, processed, or identified tandem MS (MS/MS or MS^2^) data of NPs, in an analogous fashion to what has been achieved in genomics and proteomics research with the GenBank^®^ at the National Center for Biotechnology Information (NCBI) [55] and the UniProtKB [56], the open-access knowledgebase known as Global Natural Products Social Molecular Networking (GNPS, http://gnps.ucsd.edu, accessed on 20 December 2022) was presented in 2017 [57]. The spectral libraries enable unambiguous dereplication (by matching spectral features of the unknown compound(s) to curated spectral databases of reference compounds, i.e., identification of “known unknowns”) [50], variable dereplication (approximate matches to spectra of related molecules), and the identification of spectra in molecular networks. Importantly, GNPS allows for the community-driven, iterative re-annotation of reference MS/MS spectra in a wiki-like fashion, and therefore it will contribute to library improvements and eventual convergence of all curated MS/MS spectra. The visualization of molecular networks in GNPS represents each spectrum as a node, and spectrum-to-spectrum alignments as edges (connections) between nodes.

## 3. Linking Chemical Diversity of Secondary Metabolites to Biosynthetic Gene Clusters

Secondary metabolites can be considered genetically encoded small molecules that play a variety of roles in cell biology and therefore have the potential to become chemical probes or drug leads. Their identification and characterization can benefit from a growing number of databases and genomics-based computational tools that have been compiled and hyperlinked at the Secondary Metabolite Bioinformatics Portal (SMBP (http://www.secondarymetabolites.org/, accessed on 20 December 2022) website [58]. Inherent limitations related to their low production and difficult detection, and also high rediscovery rates, can be addressed, at least in part, by searching for BGCs in genomic data and unveiling their (sometimes cryptic) metabolic potential [59]. However, the highly repetitive nature of the associated genes creates major challenges for accurate sequence assembly and analysis, hence the need for new bioinformatic tools. An example is the Natural Product Domain Seeker (NaPDoS) web service (https://npdomainseeker.sdsc.edu/napdos2/, accessed on 20 December 2022), which provides an automated method to assess the secondary metabolite biosynthetic gene diversity and novelty of strains or environments. NaPDoS analyses are based on the phylogenetic relationships of sequence tags derived from genes encoding PKS and NRPS, respectively. The sequence tags correspond to PKS-derived KS domains and NRPS-derived condensation (C) domains and are compared to an internal database of experimentally characterized biosynthetic genes, so that genes associated with uncharacterized biochemistry can be identified [60]. The latest update (NaPDoS2) greatly expands the taxonomic and functional diversity represented in the webtool database and allows larger datasets to be analyzed. Importantly, NaPDoS2 can be used to detect genes involved in the biosynthesis of specific structural classes or new biosynthetic mechanisms, and also to predict biosynthetic potential [61].

The key role of marine microbial symbionts of invertebrates in MNP biosynthesis has been increasingly recognized [62] and “genome mining” (i.e., the exploitation of genomic information for the discovery of biosynthetic pathways) [63] provides unique opportunities for (i) the identification of yet undisclosed specialized metabolites [64] and their chemical variants [63]; (ii) the genetic engineering of BGCs to obtain novel “unnatural” NPs [65]; and (iii) the heterologous expression of secondary metabolic pathways that remain silent or are poorly expressed in the absence of a specific trigger or elicitor [66]. In fact, the results of a variety of genome sequencing projects have unveiled the metabolic diversity of microorganisms (which may be overlooked under standard fermentation and detection conditions) and their tremendous biosynthetic potential. Furthermore, studies on the evolutionary history of BGCs in relation to that of the bacteria harboring them (“comparative genomics”) beautifully illustrate the mechanisms by which chemical diversity is created in nature and how some NPs represent ecotype-defining traits while others appear selectively neutral [67].

Novel algorithms have been devised to systematically identify BGCs in microbial genomic sequences [12,63,68]. A network analysis of the predicted BGCs in Proteobacteria (aka Pseudomonadota, a major phylum of Gram-negative bacteria) has revealed large gene cluster families, and the experimental characterization of the most prominent one revealed two subfamilies consisting of hundreds of BGCs encoding the biochemical machinery for the synthesis of a series of remarkably conserved lipids with an aryl head group conjugated to a polyene tail (i.e., aryl polyenes) that are likely to play important roles in Gram-negative cell biology [17]. The systematic study of BGCs in Actinobacteria (actinomycetes mainly associated to sponges in marine habitats) is complicated by numerous repetitive motifs. By combining several metrics, a method for the global classification of these gene clusters into families (GCFs) has been developed, and the biosynthetic capacity of the resulting GCF network has been validated in hundreds of strains by correlating confident MS detection of known NPs with the presence or absence of their established BGCs [69].

The Minimum Information about a Biosynthetic Gene cluster (MIBiG, https://mibig.secondarymetabolites.org/, accessed on 20 December 2022) specification is a data standard that facilitates the consistent and systematic deposition and retrieval of metadata on BGCs and their molecular products [70]. MIBiG is a Genomic Standards Consortium project that builds on the Minimum Information about any Sequence (MIxS) framework to (i) identify which genes are responsible for the biosynthesis of which chemical moieties, thus systematically connecting genes and chemistry; (ii) understand the natural genetic diversity of BGCs within their environmental and ecological context; and (iii) develop an evidence-based parts registry for engineering biosynthetic pathways and gene clusters through synthetic biology. The MIBiG standard contains dedicated class-specific checklists for gene clusters encoding pathways to produce alkaloids, saccharides, terpenes, polyketides, NRPs, and RiPPs [20].

Natural antimicrobial peptides (AMPs) have been found not only in marine fish [71,72] but also in marine invertebrates [73,74] as major components of their innate host defense systems. The Antimicrobial Peptide Database (APD, https://aps.unmc.edu/, accessed on 20 December 2022), online since 2003 and last updated in June 2022 [75], defines four unified classes of AMPs on the basis of the polypeptide chain’s connection patterns: (I) linear polypeptide chains (e.g., cathelicidins) [76]; (II) sidechain-linked peptides, such as disulfide-containing defensins and lantibiotics (i.e., lanthionine-containing antibiotics, e.g., microbisporicin, produced by the soil actinomycete *Microbispora corallina* [77] and mathermycin from the marine actinomycete *Marinactinospora thermotolerans* [78]); (III) polypeptide chains with side chain to backbone connection (e.g., bacterial lassos and fusaricidins); and (IV) circular peptides with a seamless backbone, i.e., N- and C-termini linked by a peptide bond (e.g., plant cyclotides and animal θ-defensins) [79]. The manually curated Database of Antimicrobial Activity and Structure of Peptides (DBAASP, http://dbaasp.org, accessed on 20 December 2022) provides detailed information (including chemical structure and activity against specific targets) on experimentally tested peptides (both natural and synthetic) that have shown antimicrobial activity as monomers, multimers, or multi-peptides [80]. The Collection of Antimicrobial Peptides (CAMP), CAMPSign, and ClassAMP are open-access resources that have been developed to advance our current understanding of AMPs, from N- and C-terminal modifications and the presence of unusual amino acids to 3D structures thorough family-specific signatures that facilitate AMP identification and classification as antibacterial, antifungal, or antiviral [81,82]. Synthetic AMPs are substantially enriched in residues with physicochemical properties known to be critical for antimicrobial activity, such as high α-helical propensity, positive charge, and hydrophobicity.

The Natural Products Atlas [83] was created as an open-access centralized knowledgebase encompassing ~25,000 microbially produced NPs using a combination of manual curation and automated data mining approaches, and was developed as a community-supported resource under findable, accessible, interoperable, and reusable (FAIR) [84] principles. It contains referenced data for molecular structure, source organism, isolation, total synthesis, and instances of structural reassignment for compounds of bacterial, fungal, and cyanobacterial origin. Its associated web interface (https://www.npatlas.org, v. 2.3.0, accessed on 20 December 2022) allows users to search by structure, substructure, and physical properties, as well as to explore the chemical space of these NPs from a variety of perspectives. The NP Atlas is integrated with other NP databases, including the MIBiG repository and the GNPS platform cited above. The NP Atlas was recently updated [19] and currently embodies (i) >32,000 compounds; (ii) a full RESTful (REST is an acronym for REpresentational State Transfer and an architectural style for distributed hypermedia systems) application programming interface (API); (iii) full taxonomic descriptions for all microbial taxa; (iv) integrated data from external resources, including CyanoMetDB (https://www.eawag.ch/en/department/uchem/projects/cyanometdb/, accessed on 20 December 2022), a comprehensive public database of secondary metabolites from cyanobacteria (aka “blue-green algae”) [85]; and (v) chemical ontology terms from both ClassyFire [86] (see below) and NPClassifier (a deep-learning tool for the automated structural classification of NPs from their counted Morgan fingerprints) [87].

Finally, more than seven terabases of metagenomic data from samples collected in epipelagic and mesopelagic water locations across the globe by the *Tara* (https://fondationtaraocean.org/en/foundation/, accessed on 20 December 2022) Oceans project have been used to generate an ocean microbial reference gene catalog (http://ocean-microbiome.embl.de/companion.html, accessed on 20 December 2022) with >40 million nonredundant sequences from viruses, prokaryotes, and picoeukaryotes. Remarkably, almost three quarters of ocean microbial core functionality is shared with the human gut microbiome, and epipelagic community composition was found to be mostly driven by water temperature rather than geography or any other environmental factor [88]. A more recent analysis of 214 metagenome-assembled genomes (MAGs) recovered from the polar seawater microbiomes revealed strains that are prevalent in the polar regions while nearly undetectable in temperate seawater [89].

## 4. Classification and Chemoinformatic Analyses of Natural Products

The long-established Gene Ontology (GO) resource [90,91] describes our knowledge of the “universe” of biology with respect to (i) molecular functions, (ii) cellular locations, and (iii) biological processes of gene products, in terms of a dynamic, controlled vocabulary that can be applied to prokaryotes and eukaryotes, as well as to single and multicellular organisms. Along the same vein, a standardized and purely structure-based chemical ontology (ChemOnt) was recently developed to automatically assign over 77 million compounds to a taxonomy consisting of >4800 different categories by means of a computer program named ClassyFire (http://classyfire.wishartlab.com/, accessed on 20 December 2022) that is freely accessible as a web server [86]. This new taxonomy for chemical substances consists of up to 11 different levels (kingdom, superclass, class, subclass, etc.), with each of the categories defined by unambiguous, computable structural rules. 

As a follow-on, the Chemical Functional Ontology (ChemFOnt), another FAIR-compliant, web-enabled resource (https://www.chemfont.ca, accessed on 20 December 2022), describes the functions and actions of >341,000 biologically important chemical substances, including primary and secondary metabolites, as well as drugs and NPs. The functional hierarchy within ChemFOnt consists of four functional “aspects” (physiological effect; disposition; process; and role), which are subdivided into twelve functional categories (health effects and organoleptic effects; sources, biological locations, and routes of exposure; environmental, natural, and industrial processes; adverse biological roles, normal biological roles, environmental roles, and industrial applications) and a total of >170,000 functional terms. At the time of publishing, ChemFOnt contained almost four million protein-chemical relationships and more than ten million chemical-functional relationships that can be adopted by other databases and software tools and be of utility not only to general chemists but also to researchers involved in genomics, metagenomics, proteomics, and metabolomics [92].

NPs are the result of nature’s exploration of biologically relevant chemical space through eons of evolutionary time, hence their high diversity regarding atom connectivity and functional groups. Because they cover a broad range of sizes, 3D structures, and physicochemical properties that can be related to drug-likeness (including favorable ADME characteristics), NPs are considered not only as potential drugs, but also as an invaluable source of chemical inspiration for the development of new bioactive small molecules useful in chemical biology and medicinal chemistry research. The structural diversity of drugs was early assessed by making use of shape description methods and grouping the atoms of each drug molecule into ring, linker, framework (or scaffold) [93], and side chain [94]. A methodology that calculated the NP-likeness score—a Bayesian measure of similarity with respect to the structural space covered by NPs—proved capable of efficiently separating NPs from synthetic (i.e., man-made) molecules in a cross-validation experiment [95]. Nevertheless, rule-based procedures applied to the automated assignment of NPs to different classes, such as alkaloids, steroids, and flavonoids, have unveiled database-dependent differences in the coverage of chemical space [96]. Beyond that, several cheminformatics techniques have been used to analyze NPs and decompose them into fragments in the belief that their unique substructural features and chemical properties are likely to be optimized for protein recognition and enzyme inhibition. A recent cheminformatic analysis of the structural and physicochemical properties of NP-based drugs in comparison to top-selling brand-name synthetic drugs revealed that macrocycles occupied distinctive and relatively underpopulated regions of chemical space, while chemical probes largely overlapped with synthetic drugs [97]. 

Ideally, molecular diversity in drug discovery efforts should be focused on what is usually considered drug-like chemical space (aka “drug space”), which may (or may not) fully comply with Lipinski’s “rule of five” [98]. A pioneering initiative to map this space made use of 72 descriptors accounting for size, lipophilicity (calculated log P_o/w_), polarizability, charge, flexibility (number of nonterminal rotatable bonds), rigidity (total number of rings and rigid bonds), and hydrogen bonding abilities for a set of ~400 compounds encompassing both representative drugs (“core structures”) and a number of “satellite molecules” intentionally placed outside of the drug space (i.e., possessing extreme values in one or several of the desired properties, while containing drug-like chemical fragments). By means of principal component analysis (PCA) and projections to latent structures (PLS) it was possible, after some iterations that involved the inclusion of additional randomly selected active molecules, to extract map coordinates in the form of *t*-score values and construct a chemical global positioning system (ChemGPS) [99]. The ChemGPS scores were found to describe well the latent structures extracted with PCA from a large set of compounds and appeared to be suitable for comparing multiple libraries and for keeping track of previously explored regions of chemical space. Later work (largely based on cyclooxygenase 1 and/or cyclooxygenase 2 (COX-1/2) inhibition) proposed an expansion of ChemGPS to better cover space for NPs, giving birth to ChemGPS-NP [100], which was further tuned for the improved handling of the chemical diversity encountered in NP research with a view to increasing the probability of hit identification [101]. The public ChemGPS-NP Web tool (http://chemgps.bmc.uu.se/, accessed on 20 December 2022) was then developed to allow for the exploration of NPs by navigating in a consistent 8-dimensional global map of structural characteristics built by means of PCA [102].

Following a different philosophy to chart the known chemical space explored by nature, the structural classification of natural products (SCONP) was devised to accomplish a hierarchical grouping of the scaffolds present in ~170,000 entries from the DNP by establishing parent–child relationships between them and arranging the scaffolds in a tree-like fashion [103]. Some previous processing was necessary that included structure cleansing (i.e., separation from accompanying molecules) and deglycosylation (in the case of glycosides whose active component is the aglycon part). Unfortunately, stereochemistry could not be considered in this early cheminformatic analysis so that the different possible configurations of the NP scaffolds had to be treated as being equivalent. The conversion of the resulting NP scaffolds to SMILES (simplified molecular-input line-entry system) strings [104] allowed for the comparison with those of standard synthetic molecules represented by over 10 million drug-like commercially available samples from the ZINC database [41]. This analysis revealed interesting differences not only between natural and synthetic (i.e., man-made) molecules, but also between scaffolds originating from distinct classes of organisms, i.e., plants, bacteria, and fungi. Visual comparisons of the respective structural features were effectively displayed by plotting the scaffolds according to their frequency distributions [105]. Moreover, a flexible analytics framework named Scaffold Hunter (https://scaffoldhunter.sourceforge.net/, accessed on 20 December 2022) generates and enables the visualization of virtual scaffold trees in bioactive compound collections that easily allow for the identification of new starting points for the design and synthesis of biology-oriented small molecule libraries [106]. Interestingly, a recent cluster analysis of chemical fingerprints and molecular scaffolds of >55,000 compounds reportedly isolated from marine and terrestrial microorganisms showed that three quarters of the MNPs are closely related to compounds isolated from their terrestrial counterparts [107].

The cheminformatic deconstruction of hundreds of thousands of NPs has allowed for the definition of thousands of fragment groups that represent a large portion of the chemical space defined by NPs and may guide the synthesis of “non-natural” NPs or pseudo-NPs, that is, molecules made in the lab that contain at least some of the structural features present in NPs but have not yet been found in living organisms [108]. In this regard, we must bear in mind that the prototype “antimetabolite” 6-thioguanine, which was synthesized in 1955 by Nobel Prize winners Elion and Hitchings [109], was found in 2013 to be biosynthetically produced by *Erwinia amylovora*, the bacteria responsible for fire blight pathogenesis in apple and pear trees [110]. In fact, a recent cheminformatic analysis revealed that a significant portion of biologically active synthetic compounds can be regarded as pseudo-NPs and, as such, the result of human-directed “chemical evolution” of NP structure [111]. Once again, humans imitate nature by (i) performing atom/group replacement and/or decorating with novel fragments what are thought to be privileged scaffolds for bioactivity [112,113,114]; or (ii) combining fragment-sized NPs and/or NP fragments to provide “hybrid NPs” [115].

Historically, the total synthesis of NPs followed by derivative synthesis (“active analogue approach” or “analogue-oriented synthesis” [116,117]) and semisynthetic procedures aimed at modifying the chemical structure of complex fermentation products have enabled a deeper understanding of structure–activity relationships (SAR). In contrast, the de novo combination of NP fragments in unique arrangements, often by virtue of innovative strategies such as “diversity-oriented synthesis” [118,119], “target-oriented and diversity-oriented organic synthesis” [120], and “synthesis-informed design” [121], has been shown to generate focused NP-like libraries containing compounds endowed with bioactivities unrelated to those of the guiding NP(s) [122,123,124]. Examples of successful workflows of pseudo-NP design and development are “biology-oriented synthesis” [114,125] and “pharmacophore-directed retrosynthesis” [126]. In applying the latter approach, a key first step is to elaborate a tentative pharmacophore, i.e., “an ensemble of steric and electronic features that is necessary to ensure the optimal supramolecular interactions with a specific biological target and to trigger (or block) its biological response”, as defined by the International Union of Pure and Applied Chemistry (IUPAC) [127], and then devise a retrosynthetic procedure that ensures that the proposed pharmacophore is present in multiple intermediates of increasing complexity, ultimately leading to the NP. An important goal of these synthetic approaches is to find structurally simplified and optimized derivatives with lower molecular weights that can overcome commonly observed limitations, such as poor oral absorption, short half-life, and low blood–brain barrier permeability.

## 5. Linking NPs to Their Targets: Computational Methodologies for Building Global Networks

The popular term “druggable genome” [128] refers to the genes (or, more appropriately, gene products) that are known or predicted to interact with drugs, ideally resulting in a therapeutic benefit. Although drugs are intended to be selective (i.e., have high affinity for one single target), it is not uncommon for many molecules to bind to more than one protein, giving rise to polypharmacology and side effects. Due to the fact that many drug-target combinations are theoretically possible, the computational exploration of possible interactions can help identify potential targets.

Because the systematic identification of drug targets for NPs, regardless of their origin, using a battery of experimental binding or affinity assays, is both costly and time-consuming, a substantial amount of effort has gone into devising in silico tools that allow for the construction of global networks that connect active compounds to their cellular targets. It is expected that, by using these methods, the resulting system’s pharmacology infrastructure will help to predict new drug targets for pharmacologically uncharacterized NPs and identify secondary targets (off-targets) that can aid in the rationalization of side effects of known molecules [129]. The Drug-Gene Interaction Database (DGIdb 4.0, https://www.dgidb.org/, accessed on 20 December 2022) provides information on drug-gene interactions and druggable gene products collected from publications, databases, and other web sites [130]. The latest update mostly focused on (i) the integration with crowdsourced efforts (e.g., Wikidata) to facilitate term normalization and with the open-data web platform Drug Target Commons (https://dataverse.harvard.edu/dataverse/dtc2tdc, accessed on 20 December 2022) [131] to enable the upload of community-contributed interaction data; and (ii) export to a Network Data Exchange (NDEx) infrastructure [132] for storing, sharing and publishing biological network knowledge. The tool named substructure-drug-target network-based inference (SDTNBI) was devised to prioritize potential targets for old drugs (“drug repositioning”), failed drugs, and new chemical entities by bridging the gap between new chemical entities and known drug-target interactions (DTIs) [133]. A later modification (wSDTNBI) [134] uses weighted DTI networks, whose edge weights are correlated with binding affinities, and network-based VS, which does not rely on the receptors’ 3D structures [135]. The publicly available SwissTargetPrediction web server (http://www.swisstargetprediction.ch, accessed on 20 December 2022) [136] also attempts to predict the most likely target(s) (in mice, rats, or human beings) for a SMILES-defined input molecule by using a computational method that combines different measures of similarities (both in 2D chemical structure and in 3D molecular shape) with known ligands [137]. All of these approaches, together with highly efficient receptor-based ligand docking [138], can be useful to narrow down the number of potential targets, but strict experimental confirmation and validation are needed [139,140].

The attention initially drawn [141] to certain synthetic molecules that were responsible for disproportionate percentages of hits in enzyme-based bioassays but, on closer inspection, turned out to be false actives and therefore nonprogressible hits, leading to the PAINS acronym (Pan Assay INterference compoundS) [142], was later extended to NPs [143]. As a result, some NPs have been designated as “invalid metabolic panaceas” and the concept of “residual complexity” (http://go.uic.edu/residualcomplexity, accessed on 20 December 2022) has emerged [144]. Nowadays, compounds with a PAINS chemotype can be recognized and excluded from bioassays by the judicious use of electronic substructure filters [145] and machine learning approaches [146] (e.g., Hit Dexter, https://nerdd.univie.ac.at/hitdexter3/, accessed on 20 December 2022).

Because the best link connecting NPs to their targets is arguably the experimentally determined 3D structure of the respective complexes, in the following section, I will provide some examples of MNPs and synthetic analogues that were selected on the basis of chemical novelty and submicromolar inhibition data, preferably supported by structural evidence of complex formation with pharmacologically relevant enzyme targets.

## 6. Selected Examples of MNPs Acting as Enzyme Inhibitors

The road from the research laboratory to the drug pipeline is long and winding. Quite often, molecules originally assayed for one biological activity end up showing promise for another unintended indication, either fortuitously or by following one of the computational approaches outlined in the previous sections.

Bengamides A and B (Figure 2) were first described as heterocyclic anthelmintics naturally present in the sponge *Jaspis cf. coriacea* [147], and later on, not only in other sponges from many biogeographic sites, but also in the terrestrial Gram-negative bacterium *Myxococcus virescens*. Decades of further research have shown that methionine aminopeptidases MetAP1 and MetAP2 (essential metalloenzymes that remove the initiator amino-terminal methionine from nascent proteins) are molecular targets for bengamides, which also display notable antiproliferative and antiangiogenic properties [148,149]. In fact, a synthetic analogue of bengamide B, LAF389, was the subject of a phase I anticancer clinical trial that, unfortunately, demonstrated no objective responses and also the occurrence of unanticipated cardiovascular events. The high-resolution 3D structures of both human MetAP1 and MetAP2 enzymes in complex with bengamide derivatives, including LAF389 (PDB entry 1QZY) [150], have been solved [151] and show these compounds bound in a manner that mimics the binding of peptide substrates, with three key hydroxyl groups on the inhibitor coordinating the di-Co(II) center in the enzyme active site. Renewed interest in bengamides is currently focused on their antibacterial activities against various drug-resistant *Mycobacterium tuberculosis* [152] and *Staphylococcus aureus* strains [153]. Incidentally, the mycotoxin fumagillin, first isolated from *Aspergillus fumigatus* and originally studied also as an antiangiogenic agent and human MetAP2 inhibitor [154], has been widely used for more than 60 years in apiculture to control nosema disease in honey bees effectively [155] because the microsporidian *Nosema apis* lacks MetAP1 and targeting MetAP2 suppresses infection. 

Another showcase example is provided by gracilin A (Figure 3), a nor-diterpene metabolite originally isolated from the Mediterranean sponge *Spongionella gracilis* [156], that was initially reported as a potent phospholipase A_2_ (PLA_2_) inhibitor [157] and later shown to mimic the immunosuppressive effects of cyclosporin A through interaction with cyclophilin A (CypA) [158]. In a recent pharmacophore-directed retrosynthesis application, a theoretically derived pharmacophore of gracilin A was chosen as an early synthetic target. Then, sequential increases in the complexity of this minimal structure enabled SAR profiling and the identification of structurally less complex derivatives of gracilin A that displayed selectivity for mitochondrial CypD over CypA inhibition as well as significant neuroprotective and/or immunosuppressive activities [126].

The sesterterpenoids [159] are metabolites first isolated from marine sponges of the *Thorectidae* family, which includes the genera *Cacospongia*, *Fasciospongia, Luffariella*, and *Thorecta*, that often contain biologically active butenolide and hydroxybutenolide groups in their structures [160]. The anti-inflammatory activity of manoalide and luffolide (Figure 4a) was related to the inactivation of secretory PLA_2_, whereas for cacospongionolide F this biological effect was shown to involve the inhibition of the nuclear factor-κB (NF-κB) pathway as well [161]. In contrast, the related dysidiolide (Figure 4b) from the Caribbean sponge *Dysidea etherea* de Laubenfels (Dysideidae family) was the first known natural inhibitor of the cyclin-dependent kinase (CDK)-activating phosphatases cdc25A and cdc25B, with IC_50_ values in the micromolar range [162]. Later on, dysidiolide and distinctly decorated analogues prepared from *ent*-halimic acid following a classical “active analogue approach” were shown to cause stage-specific arrest of proliferating cancer cells, but again at low micromolar concentrations [163]. Because cdc25A was found to be present in the same protein structure similarity cluster (PSSC) [164] as 11β-hydroxysteroid dehydrogenase type 1 (11βHSD1, an enzyme that catalyzes the conversion of cortisone to cortisol), the SCONP-guided [103] selection of the 1,2,3,4,4*a*,5,6,7-octahydronaphthalene scaffold present in dysidiolide led to a focused compound library (Figure 4b) that showed the submicromolar inhibition of 11βHSD1 and selectivity over 11βHSD2 [165].

The bisulfide bromotyrosine- and oxime-containing derivatives psammaplin A and bisaprasin (Figure 5) were originally characterized as nanomolar inhibitors of histone deacetylases and DNA methyltransferase enzymes [166,167], but it is known today that they are used by marine sponges, such as *Pseudoceratina purpurea* and *Aplysinella rhax,* in their chemical communication [168] and quorum sensing [169] systems to prevent biofilm formation and attenuate virulence factor expression by pathogenic microorganisms.

A number of MNPs are potent inhibitors of proteases, an important drug target class in human diseases that is integrated in MEROPS (http://www.ebi.ac.uk/merops/, accessed on 20 December 2022), a database of proteolytic enzymes, their substrates and inhibitors [170]. Gallinamide A (Figure 6), a metabolite of the marine cyanobacterium *Schizothrix* sp. that originally displayed modest antimalarial activity, was subsequently reisolated and characterized as a potent and irreversible inhibitor of the human cysteine protease cathepsin L (*k_i_* = 9000 ± 260 M^−1^ s^−1^), with 8- to 320-fold greater selectivity over the closely related cathepsins V or B [171]. Docking-guided modifications to improve the binding affinity resulted in notably enhanced potency against cathepsin L (*K_i_* = 0.0937 ± 0.01 nM and *k_inact_*/*K_i_* = 8,730,000). Gallinamide and its analogs also displayed the potent inhibition of the highly homologous cruzain, an essential *Trypanosoma cruzi* cysteine protease, as well as cytotoxic activity on intracellular *T. cruzi* amastigotes [172]. Importantly, the biochemical data indicated that inhibitor potency was driven by the rate of formation of the reversible enzyme:inhibitor complex, rather than by the rate of covalent modification. The resolution of the 3D co-crystal structure of the complex formed between cruzain and gallinamide A later confirmed the proposed binding pose and revealed the expected covalent bond formed between the drug’s Michael acceptor enamide and the active site Cys25 thiol (PDB entry 7JUJ) [173].

The in vitro antiplasmodial activity of an extract of the sponge *Theonella aff. swinhoei* collected in Madagascar was ascribed, in part, to the previously known actin-binding metabolite swinholide A [174]. Further work disclosed the presence of three unusual cyclic peptides, cyclotheonellazoles A–C (Figure 6), containing six nonproteinogenic amino acids out of the eight composing units (of which the most novel were 4-propenoyl-2-tyrosylthiazole and 3-amino-4-methyl-2-oxohexanoic acid). These macrocyclic peptides are thought to be produced by hybrid PKS-NRPS enzymes from symbiotic bacteria and were found not to be active against *Plasmodium*, but instead displayed the nanomolar and subnanomolar inhibition of chymotrypsin and elastase, respectively [175]. This latter enzyme has been considered an important target to prevent acute lung injury/acute respiratory distress syndrome (ALI/ARDS) in COVID-19 patients, and the inhibition of its activity by cyclotheonellazole A has been recently shown to reduce lung edema and pathological deterioration in an ALI mouse model, comparing favorably with the clinically approved elastase inhibitor sivelestat [176].

The (Ahp)-containing cyclodepsipeptide family of cyanobacterial NPs biosynthesized by NRPS is noteworthy for the ability of many of its members (Figure 7) to inhibit several serine proteases, most notably human neutrophil elastase and kallikreins [177], by virtue of mimicking the natural substrates. The 3-amino-6-hydroxy-2-piperidone (Ahp) unit serves as the general pharmacophore, whereas the adjacent (*Z*)-2-amino-2-butenoic acid confers selectivity for elastase [178]. The depsipeptide molassamide was purified and characterized from cyanobacterial assemblages of *Dichothrix utahensis* as a new analogue of the cytostatic depsipeptide dolastatin 13 (originally isolated from the sea hare *Dolabella auricularia*) [179] that inhibited elastase and chymotrypsin at submicromolar concentrations, but not trypsin [180]. The analysis of the X-ray crystal structure of porcine elastase in complex with lyngbyastatin 7 (PDB code 4GVU) and SAR studies resulted in the synthesis of symplostatin 5, whose activity was comparable to that of sivelestat in short-term assays and more sustained in longer-term assays [181]. Complex fractionation guided by MS^2^ metabolomics (molecular networking) [182], together with HPLC, NMR, and chiral chromatography, allowed for the identification of tutuilamides A and B from *Schizothrix* sp., along with tutuilamide C from a *Coleofasciculus* sp. These novel structures (Figure 7), which are also potent elastase inhibitors, bind reversibly to this enzyme, as shown in the co-crystal structure of tutuilamide A in complex with porcine elastase (PDB code 6TH7), despite the fact that they feature an unusual vinyl chloride-containing residue. An additional hydrogen bond relative to lyngbyastatin 7 has been proposed as the element responsible for its enhanced inhibitory potency [183]. More recently, yet another family of new Ahp-cyclodepsipeptides, the rivulariapeptolides, with nanomolar potency as serine protease inhibitors, was identified from an environmental cyanobacteria community using a scalable, bioactivity-focused, native metabolomics approach [184].

The human proteasome, a multicatalytic enzyme complex that is responsible for the regulated non-lysosomal degradation of cellular proteins, gained notorious pharmacological relevance when the synthetic boron-containing bortezomib (originally developed by ProScript to treat muscle weakness and muscle loss associated with AIDS, as well as muscular dystrophy) was approved in 2003 by the FDA as Velcade^®^ (co-developed by Millennium/Takeda and Janssen-Cilag) for the treatment of relapsed/refractory multiple myeloma (MM) and mantle cell lymphoma. Carfilzomib (Figure 8), an α′,β′-epoxyketone-containing analog of the NP epoximicin—first identified in an Indian soil actinomycete strain [185]—was also approved in 2012 as Kyprolis^®^ (Onyx Pharmaceuticals) for clinical use in MM patients, in combination with lenalidomide and dexamethasone. Marizomib (aka salinosporamide A) is a structurally and pharmacologically unique MNP that contains a β-lactone-γ-lactam (Figure 8) and is produced by the marine actinomycete *Salinispora tropica*. Marizomib not only inhibits the chymotrypsin-like activity of the proteasome (via a novel mechanism involving the acylation of the O^γ^ in the N-terminal catalytic Thr residue followed by the displacement of chloride) but also those of the caspase-like and trypsin-like subunits [186]. In addition to its “pan-proteasome” pharmacodynamic activity, marizomib crosses the blood-brain barrier, and for these reasons it has been extensively studied, first preclinically [187], and then in phase I–III clinical trials, both alone and in combination. Many other MNP scaffolds continue serving as inspiration for the design and synthesis of potent 20S human proteasome inhibitors, including carmaphycins A and B (Figure 8) from a marine cyanobacterium *Symploca* species and fellutamide B, originally isolated from *Penicillium fellutanium*, a fungus found in the gastrointestinal tract of the marine fish *Apogon endekataenia* [188]. In the development of potent covalent inhibitors of the proteasome, ligand docking and binding energy calculations have highlighted the importance of the optimization of the prior noncovalent binding mode, through conformational restraints, in a pose close to that found in the transition state [189].

Protein kinases are validated drug targets because (i) kinase deregulation plays an essential role in many disease states, and (ii) many inhibitors have already shown therapeutic benefit (almost one hundred are currently approved for clinical use). This bioactivity is of broad scope and has been reported for various MNPs obtained from different sources, including bacteria and cyanobacteria, fungi, algae, soft corals, sponges, and animals [6,190]. Of note, a significant number of them were originally isolated from terrestrial sources and subsequently found in marine organisms too, and vice versa. For example, the pan-kinase inhibitor staurosporine (a pentacyclic indolo(2,3-*a*)carbazole first discovered in 1977 from the bacterium *Streptomyces staurosporeus*) was one of the early tools used to probe the cellular effects of blocking the ATP-binding pocket in different protein kinases. In 2002, 11-hydroxystaurosporine (Figure 9) was reported to be present in an ascidian *Eudistoma* species collected in Micronesia and to be a more potent inhibitor of protein kinase C than staurosporine itself [191]. Another early potent pan-kinase inhibitor is (*Z*)-hymenialdisine, which owes its name to *Hymeniacidon aldis*, the sponge where it was originally found. These and many other MNPs inspired synthetic work on analogues and novel scaffolds that paved the ground for the discovery of imatinib (Gleevec^®^), a landmark drug that has (i) significantly improved the outcomes of patients with chronic myelogenous leukemia by inhibiting the oncogenic BCR–ABL tyrosine kinase; (ii) shown remarkable clinical efficacy in the treatment of other malignancies; (iii) helped establish the concept of “targeted therapy” in the field of cancer research; (iv) fostered the concept of “precision medicine”, i.e., tailor the chemotherapeutic treatment to the unique genetic changes in an individual’s cancer cells; and (v) fuel the extremely rich and rewarding research on protein kinase inhibitors [192].

A top priority in the development of novel protein kinase inhibitors is to understand selectivity so that the tendency of one given drug to bind to other unintended kinases (off-targets) can be suppressed or attenuated. To this end, kinome-wide inhibitory selectivity profiling is necessary because small assay panels cannot provide a robust measure of selectivity [193]. Binding site similarity searches, as performed in the KinomeFEATURE (https://simtk.org/projects/kdb, accessed on 20 December 2022) [194] and KID [195] databases, along with machine learning models that map the activity profile of inhibitors across the entire human kinome [196], as exemplified by Drug Discovery Maps [197], can be of help not only to gain insight into the structural basis of kinase cross-inhibition, but also to predict the binding affinities of novel kinase inhibitors.

Cortistatin A (Figure 10a) was isolated as an antiangiogenic steroidal alkaloid from the marine sponge *Corticium simplex* and should not be confused with the somatostatin-like cortistatin neuropeptides. It consists of a 9(10→19)-*abeo*-androstane and isoquinoline skeleton and was originally shown to inhibit the proliferation of human umbilical vein endothelial cells at nanomolar concentrations [198]. It was later found that this MNP selectively inhibits the mediator-associated cyclin-dependent kinase CDK8 and disproportionately induces the upregulation of superenhancer-associated genes in acute myeloid leukemia cell lines [199]. The crystal structure of the ternary complex of CDK8 bound to cyclin C and cortistatin A (PDB code 4CRL) revealed exquisite shape complementarity between this alkaloid and the ATP-binding pocket of CDK8 (Figure 10b), with the crucial isoquinoline [200] making essential hydrogen bonding interactions with the peptide backbone.

The last example in this review is provided by sphaerimicin A (Figure 11), a complex macrocyclic uridine nucleoside derivative isolated from *Sphaerisporangium* sp. SANK60911 using a genome mining approach focused on the enzyme uridine-5′-aldehyde transaldolase [201]. Even though this is a terrestrial actinomycete, it is closely related to other marine species [202] that contain similar BGCs, hence its inclusion in this section. Sphaeromicin A exhibits nanomolar inhibitory activity on bacterial MraY, an integral membrane enzyme that catalyzes the transfer of phospho-*N*-acetylmuramyl pentapeptide from UDP-*N*-acetylmuramyl pentapeptide (Park’s nucleotide) to the phospholipid undecaprenyl phosphate during the lipid cycle of peptidoglycan biosynthesis. In an elegant example of molecular design assisted by theoretical conformational analysis and NMR data, the simplified analogues with defined stereochemistry SPM-1 and SPM-2 were synthesized [203]. The fact that SPM-1 turned out to be 54-fold more potent than SPM-2 against MraY from *Aquifex aeolicus* (MraY_AA_) revealed the importance of the conformationally restrained macrocycle for target binding, an aspect that was clarified even further when the 3D structure of the MraY_AA_:SPM-1 complex was solved by X-ray crystallography. Therefore structure-based optimization is now feasible in order to develop MraY inhibitors with the potential of becoming novel antibiotics against drug-resistant bacteria.

## 7. Conclusions and Outlook 

Computational methodologies play indispensable roles in the exploration of the vast chemical space covered by MNPs by helping, among many other tasks, to (i) elucidate their chemical composition and 3D structure; (ii) store, process, curate, and organize huge amounts of information related to source organisms, biosynthesis, and bioactivity; and (iii) connect biological activities with both molecular scaffolds and target binding sites [204]. The limits of biologically relevant chemical space for enzyme inhibitors are defined by the specific binding interactions taking place between small- and medium-sized molecules (e.g., terpenoids, alkaloids, polyketides, non-ribosomal peptides, and RiPPs) [20] and a number of selected orthosteric and allosteric pockets in macromolecular catalysts that have evolved over billions of years [205]. 

Some of the molecular entities recently found in marine microbiota can easily defy and outperform a chemist’s imagination and ingenuity, and also be endowed with unexpected, and even unprecedented, bioactivities that may inspire more synthetic creativity. A historical example is the clinically used cytarabine (aka cytosine arabinoside or arabinosyl cytosine, Ara-C), a synthetic pyrimidine nucleoside that was developed in the imitation of spongothymidine, a nucleoside originally isolated from the Caribbean sponge *Tethya crypta*. Many other analogues, however, did not follow the same fate and, in fact, the potential of MNPs as therapeutic agents for human diseases has been realized only in a few cases, which attests to the enormous difficulties of progressing many of these compounds through the drug pipeline with the final goal of demonstrating an acceptable benefit-risk balance in clinical trials and thereafter. We must be confident that the new generations of cross-disciplinary trained scientists working in community-wide networks (e.g., Ocean Medicines, https://cordis.europa.eu/project/id/690944, accessed on 20 December 2022) will overcome existing hurdles to find valuable new medicines inspired by, or based on, MNPs.

It seems clear that the integration of information from various sources, including high-throughput phenotypic screening and BGC engineering, using computational methods has revolutionized NP research and can speed up the process of discovering new biologically active molecules from marine and terrestrial sources. A major bottleneck in these efforts is to identify the macromolecular target that is responsible for the observed (or assigned) mechanism of action, a problem that is usually aggravated when dealing with complex mixtures of MNPs. The recent success in the functional characterization of several NPs and the identification of bioactive metabolites upon integrating results from untargeted metabolomics, high-content image analysis of perturbation-treated cells, and gene expression signatures [206] on a data-driven multi-platform raises the hope for the accelerated discovery of novel pharmacologically active MNPs in the near future.

## Figures and Tables

**Figure 1 marinedrugs-21-00100-f001:**
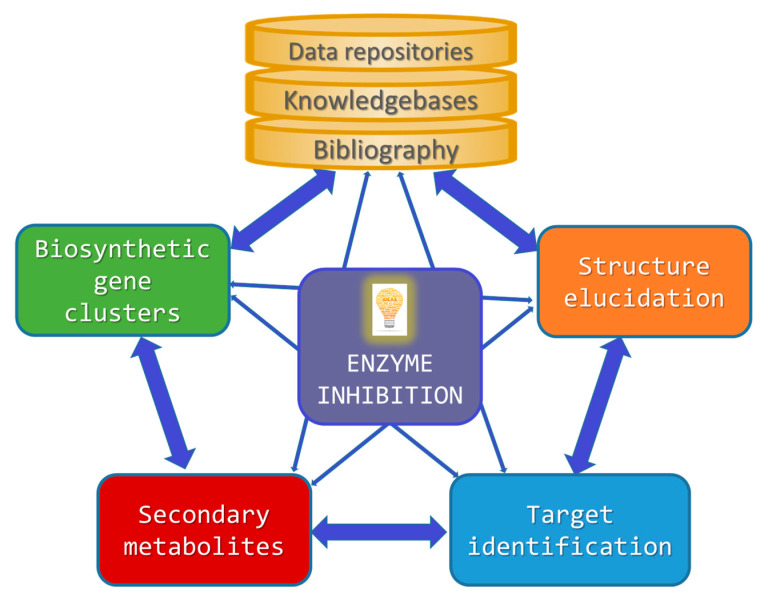
Simplified interrelationship diagram illustrating how the design/identification of enzyme inhibitors from marine sources can benefit from the use of computer-aided methods.

**Figure 2 marinedrugs-21-00100-f002:**
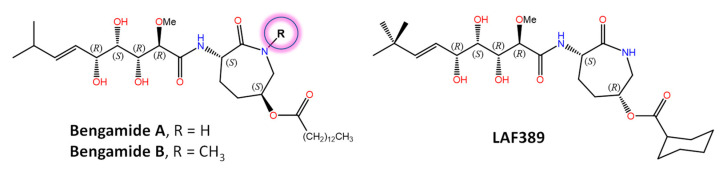
Chemical structures of naturally occurring bengamides A and B, which appear to be encoded by a mixed PKS/NRPS BGC, and the synthetic analogue LAF389.

**Figure 3 marinedrugs-21-00100-f003:**
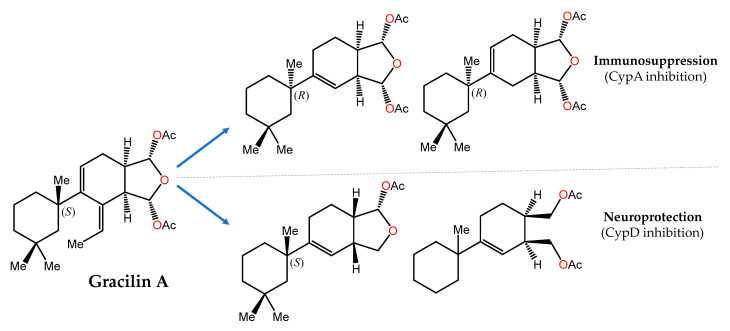
Pharmacophore-directed chemical modifications of gracillin A leading to novel compounds with distinct pharmacodynamic profiles by selective CypA vs. CypD inhibition [126].

**Figure 4 marinedrugs-21-00100-f004:**
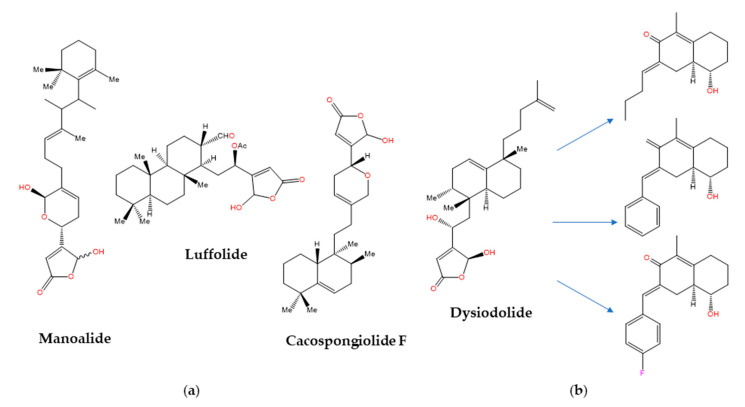
Examples of MNPs containing hydroxybutenolide moiety. (**a**) PLA_2_-inhibiting sesterterpenoids of marine origin; (**b**) SCONP-guided [103] evolution from dysidiolide to a focused library of submicromolar inhibitors of 11βHSD1 that showed some selectivity over 11βHSD2 [165].

**Figure 5 marinedrugs-21-00100-f005:**
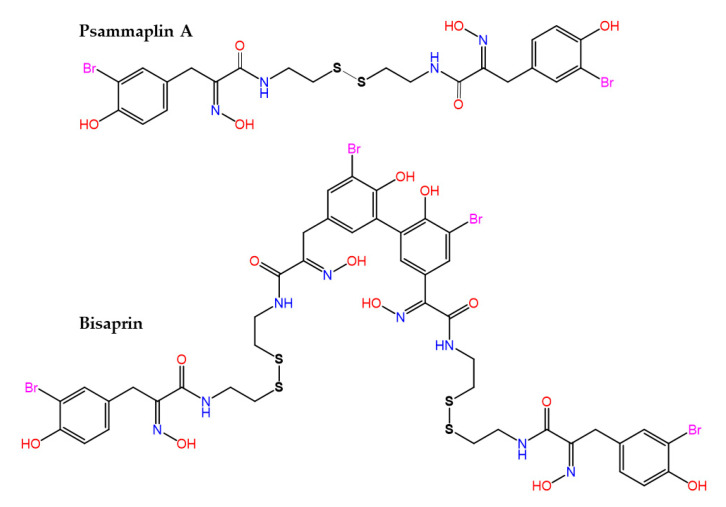
Structures of psammaplin A and its biphenylic dimer bisaprasin.

**Figure 6 marinedrugs-21-00100-f006:**
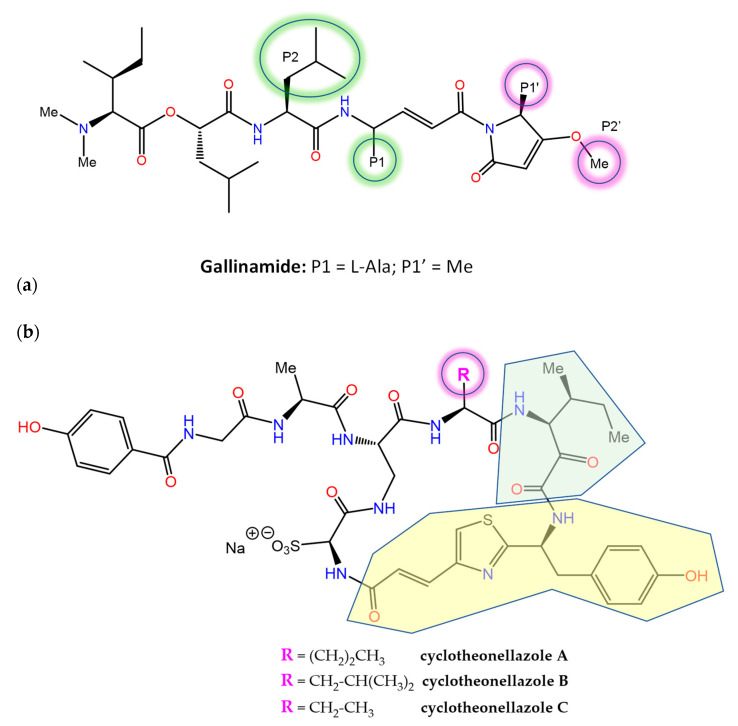
Chemical structures of protease inhibitors (**a**) gallinamide and (**b**) cyclotheonellazoles A–C. The 4-propenoyl-2-tyrosylthiazole (Ptt) and 3-amino-4-methyl-2-oxohexanoic acid (Amoha) subunits in the cyclotheonellazoles are highlighted in yellow and green, respectively.

**Figure 7 marinedrugs-21-00100-f007:**
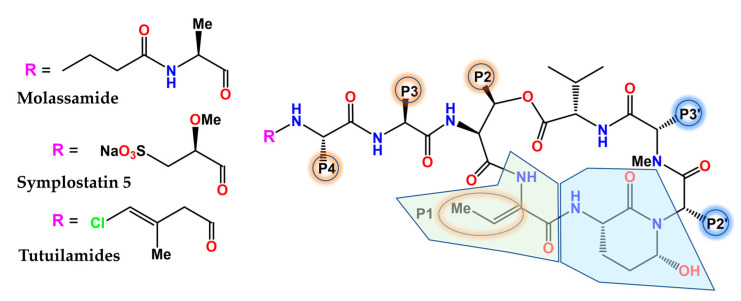
General structure of selected Ahp-cyclodepsipeptides. The 3-amino-6-hydroxy-2-piperidone (Ahp) and (*Z*)-2-amino-2-butenoic acid (Abu) subunits are highlighted in blue and green, respectively. The nature of the **P** substituents varies among family members and synthetic analogues.

**Figure 8 marinedrugs-21-00100-f008:**
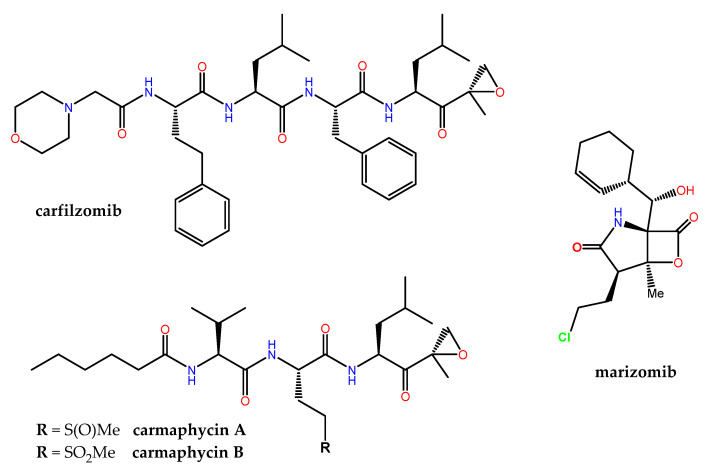
Chemical structure of selected proteasome inhibitors: carfilzomib (clinically approved), marizomib (in clinical trials), and carmaphycins A and B (investigational).

**Figure 9 marinedrugs-21-00100-f009:**
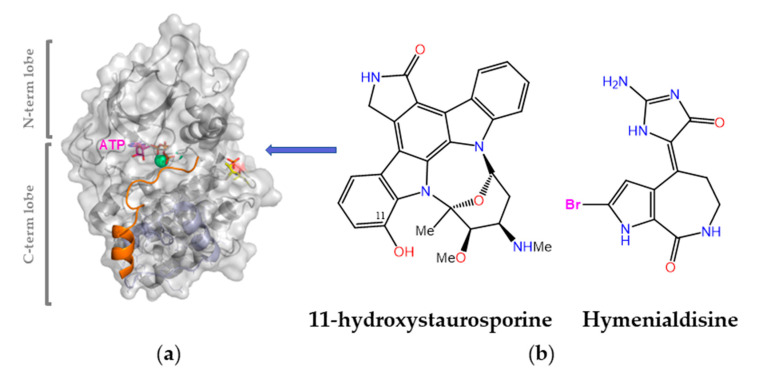
Early small-molecule protein kinase inhibitors of marine origin. (**a**) A representative protein kinase structure (PDB code 3FJQ) showing a bound peptide inhibitor (orange) and an ATP molecule (purple) plus two Mn^2+^ ions (green spheres) in the crevice that separates the N- and C-terminal lobes. The C-terminal lobe (bottom) contains the activation sites (T196 and phosphorylated T198 in yellow) and substrate-binding site (residues 230–260 in light blue). (**b**) Chemical structures of two early pan-kinase inhibitors.

**Figure 10 marinedrugs-21-00100-f010:**
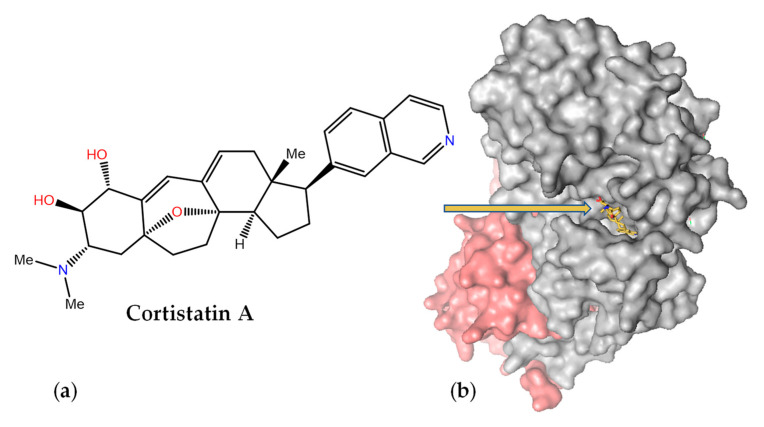
Inhibition of mediator-associated cyclin-dependent kinase CDK8 by cortistatin A. (**a**) Chemical structure of cortistatin A. (**b**) Schematic view of the complex between cyclin C (pink) and CDK8 (grey) showing cortistatin A (yellow sticks) bound inside the ATP-binding pocket of CDK8 (PDB code 4CRL).

**Figure 11 marinedrugs-21-00100-f011:**
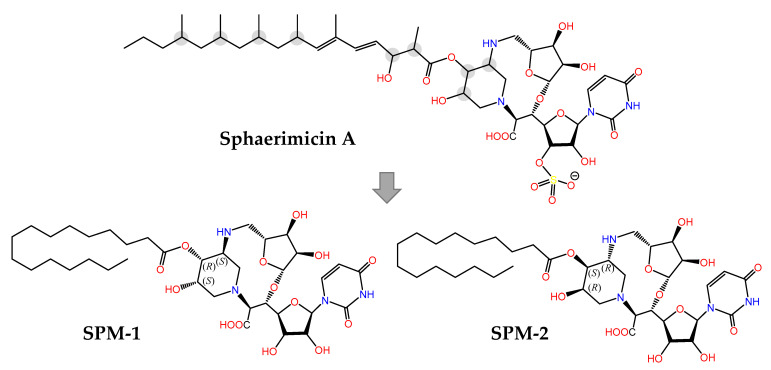
Chemical structures of the naturally occurring sphaerimicin A (containing the undetermined stereogenic centers circled in grey) and synthetic simplified analogues SPM-1 and SPM-2 with defined stereochemistry, which is critical for inhibitory potency. The X-ray crystal structure of the MraY:SPM-1 complex has recently been determined (PDB code 8CXR) [203].

## Data Availability

Data sharing is not applicable.

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
