# Peer review of "Computational Approaches to Enzyme Inhibition by Marine Natural Products in the Search for New Drugs"

_marinedrugs, 2023, doi:10.3390/md21020100_

Round 1

Reviewer 1 Report

This study contributed the extensive knowledge about the use of computational biology in term of directing particular enzyme inhibition. Manuscript is well written with clearly presented purpose and comparative analytic approaches. Specific point can be addressed in this study which is to state multitarget efficacy of marine natural product and their respective consequences on different metabolic pathways. Some minor correction needed to improve the manuscript such as

·  Compile the manuscript according to journal’s format (Introduction, heading, subheadings, conclusion etc).

·        Revise Outlook with Conclusions and Outlook

·         Add reference in line number 64

·         Add reference in line number 106

·         Add reference in line number 161

·         Review reference style of journal (remove “et al”) 

Author Response

  • The manuscript is compiled according to journal’s format even though Overview substitutes for Introduction and Conclusions and Outlook for Conclusions. The subheadings refer to the section contents ti improve readibility.
  • No extra references were needed, in my view, in lines 64, 106 and 161.
  • According to MDPI's style in EndNote, “et al” is to be used when the number of authors is greater than 10.

Reviewer 2 Report

Overall, the author has chosen an interesting and enjoyable way of tracing the line of reasoning, presenting marine natural products in a promising and attractive way, following the interface of biosynthesis and genomics. Then it presents the importance of informatics and computational processes, and ends with the pharmacological part, presenting the targets (enzymes).

 In section 1, the author chose adequate and (mostly) updated references to base his sentences and considerations. However, I would like to suggest adding a reference to the paragraph ending in "...all living species" (line 33) and adding or replacing reference 3 with a more current one.   I believe there was a certain lack of connection between the three themes all long the overview. I suggest adding a simple paragraph at the end that connects marine natural products, bioinformatics tools and pharmacological potential.   Section 2 is very well written, complete, drawn from the natural products and marine natural products databases. The author still makes pertinent considerations about the analytical techniques of detection and identification of compounds, showing examples and tools. As a suggestion and referring to what was said in the section (lines 201 to 205), it would be interesting to add information (if any) on platforms or databases for the dereplication of natural products from NMR and UV data. This additional could be done at the end of the section, as well as the author cited the MS/MS dereplication platforms.   Section 3 is quite consistent with the review proposal, since there has been great progress in the last decade on the construction of tools that correlate the biosynthetic potential of microorganisms (especially marine ones) with the production of secondary metabolites. The author presented the main platforms and database in an interesting and objective way. The examples were well applied.   In section 4, the author describes classification tools and chemoinformatics analysis of natural products. Forwarding its trajectory through the article, it continues to bring valuable information about the characterization and applicability of natural products.   In section 5, the author reaches his point, in which he presents computational platforms and methodologies that help in the discovery and elucidation of targets for natural products.   In section 6, the author shows some examples of marine natural products that act on different types of enzymes. The examples were well chosen, although very focused on sponges and cyanobacteria. I believe that this can be offset by the enzymatic diversity chosen and also by the different classes of natural products presented.   In section 7, at the end, the author makes his final considerations citing one more example (Ara-C), and  pointing to chemical synthesis as a direction to be followed. I must confess that I expected a little more from this final part. It would be interesting for the author to point out, if it is coherent, which types of platforms (of the several mentioned throughout the text, or all of them?) can be integrated to bring more speed and richness of information in - just one place -. Or if there are areas related to research on marine natural products that still need organization for the establishment of integrative data platforms that help the scientific community. I'd like to see a little more author input on the conclusions and perspectives.

Author Response

In section 1, a reference has been added to the paragraph ending in "...all living species" (line 33). Unfortunately, ref. 3 cannot be replaced with a more current one because I could not find one. However, the sentence has been rewritten to make the most of the footnote referrring to the current pipeline.

To help linking the three broad and diverse subjects covered in the overview, a new figure (Figure 1) has been added. This was also suggested by the editor at an earlier stage.

A simple paragraph has been added at the end that connects marine natural products, bioinformatics tools and pharmacological potential.  

Section 2: there is no information (to the best of my knowledge) on platforms or databases for the dereplication of natural products from NMR and UV data.

In section 6, the focus was not on particular organisms as sources of MNPs, but rather on data quality and relevance of the findings, ideally supported by 3D structure determination.

In section 7, the sentence citing Ara-C as a successful example has been extended and further qualified.

Finally, a last paragraph on platform integration and hope for the future (from the author's viewpoint) has been included.

I thank the reviewer for his/her careful reading and valubale suggestions.

Reviewer 3 Report

Gago's manuscript (marinedrugs-2143242) summarized the advanced computational approaches to enzyme inhibition by MNPs for drug discovery, especially focusing on five categories: virtual NP databases, biosynthetic gene clusters, chemoinformatic analyses, global networks, and examples. After reading the paper, I found the review well summarized, with plenty of information that is expected to be valuable for the readers. But I still wonder about the basic principle used to select the examples of MNPs acting as enzyme inhibitors in this review. Can it be interpreted in detail?

Overall, I recommended consideration for publication within "Marine Drugs" after minor revisions.

Author Response

The basic principle used to select the examples of MNPs acting as enzyme inhibitors in this review is now explained as follows: "Because the best link connecting NPs to their targets is arguably the experimentally determined 3D structure of the respective complexes, in the following section I will provide some examples of MNPs and synthetic analogues that were selected on the basis of chemical novelty and submicromolar inhibition data, preferably supported by structural evidence of complex formation with pharmacologically relevant enzyme targets."

I would like to thank the reviewer for bringing this issue to my attention.